# Solute Transport through Mitochondrial Porins In Vitro and In Vivo

**DOI:** 10.3390/biom14030303

**Published:** 2024-03-04

**Authors:** Roland Benz

**Affiliations:** Science Faculty, Constructor University Bremen, Campus-Ring 1, 28759 Bremen, Germany; rbenz@constructor.university

**Keywords:** mitochondrial porin, VDAC, voltage dependence, peripheral kinases, lipid bilayer, pore structure, mitochondrial metabolism, cancer, apoptosis

## Abstract

Mitochondria are most likely descendants of strictly aerobic prokaryotes from the class *Alphaproteobacteria*. The mitochondrial matrix is surrounded by two membranes according to its relationship with Gram-negative bacteria. Similar to the bacterial outer membrane, the mitochondrial outer membrane acts as a molecular sieve because it also contains diffusion pores. However, it is more actively involved in mitochondrial metabolism because it plays a functional role, whereas the bacterial outer membrane has only passive sieving properties. Mitochondrial porins, also known as eukaryotic porins or voltage-dependent anion-selective channels (VDACs) control the permeability properties of the mitochondrial outer membrane. They contrast with most bacterial porins because they are voltage-dependent. They switch at relatively small transmembrane potentials of 20 to 30 mV in closed states that exhibit different permeability properties than the open state. Whereas the open state is preferentially permeable to anionic metabolites of mitochondrial metabolism, the closed states prefer cationic solutes, in particular, calcium ions. Mitochondrial porins are encoded in the nucleus, synthesized at cytoplasmatic ribosomes, and post-translationally imported through special transport systems into mitochondria. Nineteen beta strands form the beta-barrel cylinders of mitochondrial and related porins. The pores contain in addition an α-helical structure at the N-terminal end of the protein that serves as a gate for the voltage-dependence. Similarly, they bind peripheral proteins that are involved in mitochondrial function and compartment formation. This means that mitochondrial porins are localized in a strategic position to control mitochondrial metabolism. The special features of the role of mitochondrial porins in apoptosis and cancer will also be discussed in this article.

## 1. Introduction

Eukaryotic cells evolved most likely from the Last Eukaryotic Common Ancestor (LECA). The special feature of this organism is described by the symbiosis of two to three prokaryotic cells about one billion years ago [1,2,3]. With the support derived from the endosymbiont’s mitochondria and chloroplasts, the protoeukaryotic cell capable only of fermentation, derived access to oxidative phosphorylation and photosynthesis [4]. The energy-delivering endosymbiont in terms of aerobic respiration was presumably a member of *α-proteobacteria* [5,6]. This can be concluded from the homology of aerobic respiration between mitochondria and *α-proteobacteria* [6].

*α-Proteobacteria* are surrounded by two membranes [7]. The bacterial outer membrane acts as a molecular sieve for the passive diffusion of hydrophilic solutes because of the presence of pore-forming proteins, the bacterial porins [8]. Similarly, the mitochondrial outer membrane also contains pore-forming proteins that allow the passage of mitochondrial metabolites. The functional properties of this pore will be discussed in full detail in this article. The knowledge of eukaryotic porins or VDACs started in the late seventies of the last century with a study by Schein et al., who reconstituted a pore from crude extracts of *Paramecium* mitochondria into planar lipid bilayer membranes [9]. This pore was highly voltage-dependent, which will be discussed here in detail together with its consequences on mitochondrial metabolism. The protein responsible for pore formation was not known at the beginning, but very soon after, it was suggested that the pore found in the crude extracts of *Paramecium* mitochondria was also present in the mitochondrial outer membrane of rat liver mitochondria [10]. The first identification of the pore-forming protein occurred in a study of mitochondria from mung beans [11]. In that paper, it was shown that the pore-forming activity was present in fragments of the mitochondrial outer membranes of the organelles. The reconstitution of the fragments into vesicles from soybean lipids led to the permeabilization of the vesicles for low-molecular-mass carbohydrates but not for high-molecular-mass dextran [11]. Detailed investigation of the mitochondrial outer membranes of mung beans resulted in the identification of a protein with a molecular mass of about 30 kDa that was responsible for the permeability properties of the reconstituted vesicles [11]. Yeast porin was the next mitochondrial porin to be identified [12]. It was isolated as a protein with a molecular mass of 29 kDa from the outer membrane of yeast mitochondria. Combined with its identification was also the observation that it was presumably deeply buried in the mitochondrial outer membrane because it was resistant in its membrane form against the action of proteases [12]. In vitro synthesized yeast porin incorporated directly into intact yeast mitochondria, indicating that there was no leader sequence for the sorting of the protein [12]. The results obtained from biosynthesis and translation of yeast porin were supported by similar experiments with porin from *Neurospora crassa* mitochondria [13]. Porin from *N. crassa* had approximately the same molecular mass as yeast porin and was found to be voltage-dependent [12,13,14,15].

The first mitochondrial porin that was identified from mammalian mitochondria was rat liver porin [16]. It had a molecular mass of about 30–35 kDa, which was similar to the other known mitochondrial porins, and was also voltage-dependent, similar to the porins from yeast and *Paramecium* [9,15,16]. The molecular mass of rat liver porin was confirmed by other investigations [17,18]. Interesting new features of mitochondrial porins included the observation that peripheral kinases, such as hexokinase and glycerol kinase, were bound to it [19,20]. More recent research provided evidence that eukaryotic porins also play an important role in other mitochondrial features such as mitochondria-mediated apoptosis and protein translocation [21,22,23,24]. Similarly, they are presumably also involved in the response to drugs. This applies to the interaction between mitochondrial porin and the 18 kDa translocator protein (TSPO), also known as tryptophan-rich sensory protein or peripheral benzodiazepine receptor (PBR) localized in the mitochondrial outer membrane, which mediates cholesterol transport between mitochondrial membranes, cytochrome C release, and apoptosis [25,26,27]. It is also involved in porphyrin transport and stress control in mitochondria [28,29]. Mitochondrial proteins encoded in the nucleus do not exist only in eukaryotic cells. It is interesting to note that the genome of the prokaryotic pathogen *Legionella pneumophila* encodes for some proteins that show high homology to specific proteins in mitochondria, like hVDAC1 (Lpg 1974), PBR (Lpg 0211), and cyclophilin D (peptidyl prolyl isomerase D) (Lpg 1982) [30]. The role of Lpg 0211 and Lpg 1982 in *Legionella* is not well understood [29], but Lpg 1974, which serves as an outer membrane porin, forms voltage-dependent pores in lipid bilayer membranes with similar characteristics as hVDAC1 [31].

## 2. Isolation and Purification of Eukaryotic Porins

A prerequisite for the study of mitochondrial porins in reconstituted systems (lipid bilayer, lipid vesicles) is their isolation and purification from eukaryotic cells. For this, mitochondria must be isolated from eukaryotic tissue or cells by density centrifugation following disruption of cells and tissues [11,12,16]. The next step consists of the swelling and shrinking of mitochondria to release the outer membrane (MOM). This is not very efficient because the MOM is partially tightly associated with the mitochondrial inner membrane at so-called contact sites [32], which means that its isolation is combined with a considerable loss of material. Mitochondrial porins were obtained by detergent-mediated digestion of purified MOM using non-ionic detergents. Ionic detergents could not be used for their isolation because they destroyed their pore-forming activity [16,33], presumably by dissociation of the tertiary structure of the protein. Final purification of the pore-forming protein was achieved by different chromatographical steps [11,12,14,16].

A major step forward in the research of the mitochondrial outer membrane pore was the method introduced by Freitag et al. [14]. It started from whole mitochondrial membranes that were obtained from mitochondria by osmotic lysis followed by centrifugation. The total mitochondrial membranes were dissolved in non-ionic detergents followed by passing them through a dry hydroxyapatite (HTP) column [14]. Most proteins bound to the column material; only the mitochondrial porin, which was deeply buried in the detergent micelle, was not bound to the column and was found with some impurities in the eluate of the column [14]. The eluate was passed in a second step through a dry HTP/celite column in a ratio of 1:1 (*w*/*w*). Using this method, *N. crassa* porin was almost pure [14]. Later, further refinement of the purification of mitochondrial porin was possible by the method of De Pinto et al. [34]. Using this method, mitochondrial membranes were dissolved in 3% Triton X-100 using a low protein/detergent ratio and then passed only once through a dry HTP/celite column in a ratio of 2:1 (*w*/*w*) [34]. Eukaryotic porins were obtained by this procedure in high purity. It is noteworthy that this simple method was successfully used for the purification of different eukaryotic porins by the Bari/Catania group in their research into mitochondria [34,35,36,37,38]. Using this method many mitochondrial porins could be studied in reconstituted systems [33,39,40]. Similarly, the structure and function of eukaryotic porins and their interaction with different detergents could also be studied using the simple purification procedure by passing total mitochondrial membrane proteins dissolved in detergent through an HTP/celite column [41,42,43,44]. Common to all mitochondrial or eukaryotic porins known to date is their molecular mass of around 30 kDa. This suggests that they are closely related despite substantial variations in amino acids in the primary sequences [33,45,46,47].

## 3. Heterologous Expression of Mitochondrial Porins in *Escherichia coli*

The primary amino acid structure of eukaryotic porins was known quite early in the case of yeast porins and porins from *Neurospora crassa* [48,49]. For eukaryotic porins from mammals, the primary sequence was not known. However, it was quite clear that their amino acid composition was not very hydrophobic, although the pore was deeply buried in the MOM [16,48,49,50,51]. This result indicated some relationship with bacterial porins because the hydrophobicity of their amino acid distribution is close to that of soluble proteins [8]. The channels formed by bacterial porins are lined up by amphipathic β-strands and form β-barrel cylinders [7,8,33].

The sequencing of human Porin 31HL on the amino acid level allowed for the cloning and sequencing of many different mitochondrial porins from mammals followed by their heterologous expression in *Escherichia coli* [52,53,54,55,56,57]. Three different isoforms of eukaryotic porins were discovered in many different organisms [55,56,57,58]. The differences in the primary sequences of the three VDAC isoforms in mammals did not alter the organization of the three genes or the structure of the splicing sequences [47,59,60]. In addition, the primary sequences of different plant porins were evaluated [45,46,61,62,63]. Many eukaryotic genomes contain more than one gene coding for homologs of eukaryotic porins with not fully understood differences in function [54,56,58]. More than one hundred sequences of eukaryotic porins are known to date. Although the sequence identity between them is relatively low, the polypeptide length and, in particular, the electrophysiological characteristics are highly preserved [33,39,40,56,64]. This means that all eukaryotic porins studied to date are anion-selective in the open state [8,9,16,39,40]. Similarly, eukaryotic porins form voltage-dependent channels that switch to lower conductance cation selective states at voltages beginning with about 20 mV.

### Renaturation of Heterologously Expressed Eukaryotic Porins

Mitochondrial porins are similar to most mitochondrial proteins encoded by the nucleus, synthesized at cytoplasmic ribosomes, and transported post-translationally into mitochondria [65,66]. The heterologous expression of eukaryotic porins in *E. coli* offers an elegant method for the mass production of mitochondrial porins that are needed for structural studies and electrophysiology. The renaturation of the expressed proteins is possible in vitro, similar to water-soluble forms of mitochondrial porins of different organisms [67,68,69,70], although the translation of the protein into mitochondria after synthesis in vivo at cytoplasmic ribosomes has nothing to do with the folding of the in vitro protein in non-ionic detergents. Mitochondrial porins are transported post-translationally via the Tom40 complex into the intermembrane space of mitochondria and inserted via the Tob44/Sam50 protein, a member of the Omp85 family of proteins in the mitochondrial outer membrane [71,72,73]. Heterologous expression of eukaryotic porins also allows for easy mutations of single important amino acids and the deletion of stretches of amino acids within their primary structure, which helps to study the function of these amino acids in channel gating and voltage dependence [68,74,75].

Eukaryotic porins synthesized at cytoplasmic ribosomes and heterologously expressed proteins are presumably at least partially if not completely unfolded. In this form, porin is inactive in reconstitution experiments using lipid bilayers [67,68,69]. Treatment of water-soluble or expressed eukaryotic porins with non-ionic detergents in the presence of sterols results in the formation of pores with properties that cannot be distinguished from the detergent-solubilized form [68,70]. Possibly, the detergents act as some kind of chaperon together with cholesterol. Epicholesterol, which is an epimeric analog of cholesterol, on the other hand, had no influence on the renaturation of water-soluble mitochondrial porin, probably because of the different vertical localizations of hydroxyl groups in both molecules [76]. In this respect, it is interesting to note that the presence of cholesterol in a ratio of five cholesterol per one polypeptide has been detected in purified eukaryotic porin from bovine hearts using different detergents [41]. Similarly, up to five cholesterol binding sites were also detected in VDAC1 using photolabeling and other techniques [77,78]. Sterols were also necessary when the properties of mutated *N. crassa* porin were studied in lipid bilayer membranes [68]. Investigations of the renaturation process of different eukaryotic porins suggest that sterols seem to be necessary, although they may also modulate the properties of the pore-forming characteristics of plant porins [38,40,63,76,79]. However, other groups found no requirement for sterols in functional renaturation following the mass production of two isoforms of human porin (hVDAC1 and hVDAC2) and of potato VDAC36 [63,70,80]. On the other hand, ergosterol clearly interacts with the eukaryotic porin of *N. crassa* and influences the environment of aromatic amino acids within the protein dissolved in detergent [81,82], and stigmasterol seems to be important for the proper function of bean seed VDAC [83]. Similarly, sterols were found to be important for the renaturation of VDAC from pea root plastids (double-enveloped cell organelles, for instance, amyloplasts) [76]. Taken together, the contradictory results suggest that it is an open question whether sterols are important for porin function and/or only accelerate the renaturation process but are essentially not needed for the formation of functional pores. It is also possible that other factors, such as the ionic strength of the aqueous solutions are important for the functional renaturation of eukaryotic porins.

## 4. Reconstitution of Eukaryotic Porins in Liposomes

The properties of the pores formed by eukaryotic or mitochondrial porins (also known as VDACs) were studied in different model membranes, such as liposomes or lipid vesicles and planar lipid bilayers. The pore-forming properties of fragments of the outer mitochondrial membranes from rat livers and mung beans were studied first in lipid vesicles [11]. To study the pore properties, the fragments were fused with liposomes from soybean lipids. This procedure made the liposomes permeable for low-molecular-mass carbohydrates [11]. The pore-forming protein within the outer membrane fragments of mung bean mitochondria was recognized as a 30 kDa protein [11]. The reconstitution of this protein or fragments from the outer membrane of rat liver mitochondria into liposomes allowed for a rough estimate of the size of the pores formed by the corresponding eukaryotic porins. The reconstituted liposomes were loaded with radioactively labeled oligo- and polysaccharides of various sizes and were passed through a gel filtration column [11]. The cut-off of the carbohydrates retained within the liposomes was polydisperse between about 2000 and 8000 Da, which suggested an exclusion limit of approximately 4000 to 6000 Da [11]. The pores seemed to be general diffusion pores that appeared to be unspecific, which is not surprising because the selectivity of channels or pores in liposomes cannot be evaluated using charged solutes because of the generation of diffusion potentials by the asymmetric distribution of the charged solutes across a membrane. A similar exclusion limit of about 3400 Da was observed for large multi-walled liposomes made of phospholipids and mitochondrial membrane material from *N. crassa*, which suggested a diameter of the mitochondrial pore of about 4 nm [84]. Similar results were obtained from experiments with rat liver porin reconstituted into vesicles. The porin made the vesicles permeable for ^14^C-sucrose but not for high-molecular-mass ^3^H-dextran, which suggested that a specific pathway, not a leak, is introduced into the vesicles [17].

## 5. Electrophysiology of Mitochondrial Porins

Many reconstitution studies with mitochondrial porins were performed using the planar lipid bilayer technique. Basically, two different methods were used for the formation of planar bilayers. Painted lipid bilayers were formed from a solution of lipid mixtures or of pure lipids in organic solvents, preferentially in n-decane, according to the classical method of Mueller et al. [85]. The lipid solution (at a concentration of about 1% weight/volume) is painted across a circular aperture with an area of about 0.5 mm^2^ on a Teflon wall. First, a lamella is obtained, which shows Newton’s colors in reflected light. This lamella becomes black in reflected light in a short while when it is thinning because the light reflected at the front side of the thin lipid film (thickness about 5 nm) and that reflected at its back side become inexistent in the eye because of the phase jump at the front side of the bilayer and the short way of the light within the membrane. Painted membranes contain about 30% solvent [86], which makes them thicker than bilayers obtained by the folding method (thickness about 3 nm) introduced by Montal and Mueller [87]. Using this method, lipids are spread on the aqueous surfaces on both sides of a Teflon foil below a small hole (10 to 50 µm). Then, the aqueous phases are raised to eventually form a bilayer by apposition of their hydrocarbon chains across the hole [87]. Bilayer formation cannot be controlled in this case by optical means because the hole is too small; its formation must be monitored by measurement of the electrical capacity of the bilayer using a voltage clamp. This type of bilayer is very often described as solvent-free; however, measurements of the contact angle between monolayer-coated water and Teflon suggested that the boundary conditions for the formation of stable bilayers can be satisfied only when a nonpolar solvent is present [88]. This means that the formation of bilayers of this type needs the presence of alkane solvents such as hexane or hexadecane or similar materials that form a torus around the bilayer. Therefore, it may be classified as solvent-depleted but not as solvent-free [88].

Using both methods, the reconstitution of mitochondrial porin is relatively simple: Purified porin dissolved in non-ionic detergent solutions is added in small concentrations (10 ng/mL to 1 µg/mL) to the aqueous phase bathing black lipid bilayer membranes formed according to the two different methods. It must be noted here that the reconstitution of eukaryotic porins gave similar results with both types of artificial membranes. It seems that despite the difference in thickness of the two types of membranes and other putative structural differences, eukaryotic porin pores create their own environment in the lipid bilayer in such a way that their genuine properties do not depend too much on the surrounding lipids and solvents in the membrane.

### 5.1. Single-Channel Analysis of Eukaryotic Porins

The addition of mitochondrial porin at a small concentration to preformed planar bilayers resulted in a strong increase in the membrane conductance (that is, the current per unit voltage) from about 0.01 µS/cm^2^ to 100 µS/cm^2^. In general, the conductance increase after the addition of the protein was not sudden but increased strongly for about 15–20 min. After that time, the membrane conductance increased at a much slower rate. When the rate of conductance increase was relatively slow (as compared with the initial one), it was shown that for different mitochondrial porins, the membrane conductance was a linear function of the protein concentration up to porin concentrations of about 1 µg/mL [15,16,89]. At that concentration, the porin-induced conductance increased normally saturated. The conductance increase was approximately linearly dependent on the concentration of porin in the aqueous phase until saturation.

When small concentrations of mitochondrial porin were added to the aqueous phase bathing of a black lipid bilayer membrane at a high current resolution of the current amplifier, the membrane current started to increase in a stepwise fashion. This process indicated the insertion of ion-permeable channels into the membrane, as it was found for many mitochondrial porins [9,10,15,16,18,89]. Figure 1 shows the reconstitution of hVDAC1 (also known as Porin 31HL [52]) in a black lipid bilayer from diphytanoyl phosphatidylcholine/n-decane. After a short delay of time, the presence of the porin resulted in a stepwise increase in the membrane conductance at a 10 mV membrane voltage. Most of the increases were directed upward at this low transmembrane potential because the steps corresponded only to the reconstitution of single preformed eukaryotic porins into the membrane. The single-channel conductance of Porin 31HL was under these conditions about 4 nS (see Figure 2, which shows a histogram of the current fluctuations obtained with Porin 31HL). Some current fluctuations had a smaller conductance of around 2 nS. These steps presumably represent substates of hVDAC1, which is voltage-dependent [90] (see also below). The number of channels formed in a lipid bilayer membrane was dependent on time and the concentration of protein. High protein concentration resulted often in such a rapid increase in conductance that the single steps could no longer be resolved.

Similar lipid bilayer experiments were performed with many eukaryotic porins by different research groups [9,16,18,91,92,93]. The pores from mammals and other animals had an open state between 4 and 4.5 nS at small voltages. Plant porins tended to have a somewhat smaller conductance [45,93,94,95]. Table 1 shows a summary of the single-channel conductance of a variety of eukaryotic porins [89].

The measurements were performed in 1 M KCI, pH 6, if not indicated otherwise. The pores were measured at low transmembrane potentials, where almost all pores should be in their open configuration. If not indicated otherwise, the single-channel conductance of the eukaryotic porins refers to VDAC1, which is the most prominent eukaryotic porin in most organisms. The table was taken from ref. [89].

Table 2 shows the electrophysical properties of pores formed by Porin 31HL (hVDAC1) in lipid bilayer membranes. The data in Table 2 suggest that the pores formed by Porin 31HL are wide and water-filled. A similar conclusion was made for many other mitochondrial porins. This is not surprising because many solutes should be permeable through the mitochondrial outer membranes. This is also the result of single-channel measurements with salts composed of different anions and cations. Even large organic anions and cations such as Tris^+^ and Hepes^−^ were found to be permeable through the open state of mitochondrial porins (see Table 2). This agrees with permeability measurement with liposomes, where the cut-off of the carbohydrates retained within the liposomes was polydisperse between about 2000 and 8000 Da, which suggested an exclusion limit of approximately 4000 to 6000 Da [11]. The single-channel conductance of the salts in Table 2 and in KCl is a linear function of the specific conductivity of the bulk aqueous phase at small membrane potentials [90].

The solutions contained 5–10 ng/mL of Porin 31HL and less than 0.1 µg/mL of the non-ionic detergent NP 40; the pH was between 6.0 and 7.0. The membranes were made of diphytanoyl phosphatidylcholine/n-decane; T = 20 °C; and V_m_ = 10 mV. G was determined by recording at least 70 conductance steps and averaging over the right-hand maximum (see Figure 2). c is the concentration of the salt solution. Conductance data were taken from ref. [90].

### 5.2. Mitochondrial Porins Are Voltage-Gated

The current recording shown in Figure 1 demonstrates that the pores formed by Porin 31HL are mostly in the open configuration at 10 mV transmembrane potential. At higher voltages, beginning at about 15–20 mV, the number of closing events increases. They are always smaller than the initial on-steps, which indicates that the pore switches to ion-permeable substates at voltages higher than 15–20 mV. An experiment of this type is shown in Figure 3. The reconstitution of Porin 31HL (hVDAC1) in a lipid bilayer membrane is measured at a transmembrane potential of 30 mV. The insertion of the pores is indicated by large conductance steps of about 4 nS (arrows in Figure 3). The action of the voltage on the pores results in their switching to substates of different amplitudes that also do not appear to be stable and show frequent on and off behavior.

Similar experiments were also performed with membranes containing many pores formed by Porin 31HL (hVDAC). Part of an experiment of this type is shown in Figure 4. Voltages of 30 mV and −30 mV, followed by 40 and −40 mV, were applied to a diphytanoyl phosphatidylcholine/n-decane membrane containing about 50 hVDAC1 pores. In this case, the closing steps of the pores could not be resolved because of their high number in the membrane. The current decayed in a single exponential function for positive and negative voltages (see Figure 4). The results of this and additional experiments at different voltages allowed for the evaluation of the voltage dependence of Porin 31HL.

The steady-state conductance of Porin 31HL showed a bell-shaped curve as a function of the applied voltage when the conductance at a given voltage G(V_m_) divided by G_0_ at zero potential was plotted as a function of membrane voltage V_m_. Figure 5 shows the results for Porin 31HL and three different salts (KCl, K-MES, and TRIS-Cl, all at pH 7.2). The results differ considerably for the different salts, presumably because the selectivity of a pore in the open and closed configuration is different. Nevertheless, the voltage dependence of Porin 31HL is approximately the same in the different salts. The data given in Figure 5 could be analyzed by a Boltzmann distribution, as proposed by Schein et al. [9]. The ratio of the number of open, $N_{o}$, to closed channels, $N_{c}$, is calculated using the data of the bell-shaped curves show in Figure 5:(1)$$N_{o}{/N}_{c}{=(G-G}_{min}{)/(G}_{0}-G)$$
where *G* is in this equation the conductance at a given membrane potential $V_{m}$, and $G_{0}$ and $G_{min}$ are the conductance at zero voltage and very high potentials, respectively. The open to closed ratio of the channels, $N_{o}/N_{c}$, is given by a Boltzmann distribution [9,102]:(2)$$N_{o}{/N}_{c}{=exp(-nF(V}_{m}{-V}_{0})/RT)$$
where *F* (Faraday’s constant), *R* (gas constant), and *T* (absolute temperature) are standard symbols, *n* is the number of gating charges moving through the entire transmembrane potential gradient for channel gating (i.e., a measure of the strength of the interaction between the electric field and the open channel), and $V_{0}$ is the potential at which 50% of the total number of channels are in the closed configuration (i.e., $N_{o}/N_{c}$ = 1).

A semilogarithmic plot of the data given in Figure 5 for 0.5 M KCl shows that they could be fitted to a straight line with a slope of 13 mV for an e-fold change in $V_{m}$. This result suggests that the number of charges involved in the gating process is approximately two in the case of Porin 31HL (hVDAC1; see Figure 6). As pointed out above, the distribution of open and closed channels, $N_{o}/N_{c}$, follows a Boltzmann distribution. This means that Equation (2) also allows for the calculation of the energy difference for channel gating:(3)$$N_{o}{/N}_{c}{=exp[-W(V}_{m})/(RT)]$$
where $W(V_{m})$ is the voltage-dependent energy difference in one-mole channels between the open and closed states. A comparison of Equations (2) and (3) shows that $W(V_{m})=nF(V_{m}-V_{0})$. The energy needed for channel closure, $nFV_{0}$, calculated from the data in Figure 5 and Figure 6 is approximately 7.7 kJ/mol, which is not a very high energy. It is considerably below the energy of one mole of hydrogen bonds, which means that channel gating is a low-energy process.

The time constant *τ* of the single exponential relaxation process shown in Figure 4 decreases with increasing voltage. Interestingly, the time constants could be fitted to a similar formalism as given in Equation (2) for the ratio $N_{o}/N_{c}$, which means that a semilogarithmic plot of the time constants versus voltage yields a straight line [102]. The slope of this line corresponded to an e-fold decrease in the time constant *τ* for an increase in the voltage of about 13 mV under the conditions of Figure 4. This result suggested again that the number of gating charges involved in channel gating is about two, which agreed satisfactorily with the number of gating charges derived from the plot of $N_{o}/N_{c}$ (see Figure 6). The time constant of the switching of the pores from the “closed” to the “open” state could not be followed for mitochondrial porins within the time resolution of the experimental instrumentation (about 1 ms). This result indicated largely different reaction rates for the closing and opening processes of the mitochondrial pores.

The voltage dependence of mitochondrial porins from a variety of eukaryotic organisms was investigated in detail in many studies: *Paramecium* [9,102,104]; mammals including rats [16,18,36], rabbits [35], bovine [35], pigs [35], and the human brain [105]; fish including *Anguilla anguilla* [99]; plants including *Arabidopsis* [106], potatoes [45,63], peas [46], corn [46,107,108], wheat [95], and pea root plastid porin [46,69]; other organisms including *Neurospora crassa* [15], yeast [37,51,91], *Dictyostelium* [53]; and flies including *Protophormia* [101] and *Drosophila* [58,92,109]. Common to all the pores investigated in these studies is that the mitochondrial porins of all these eukaryotes formed high-conducting channels in reconstituted systems. They were all in their open configuration at small transmembrane voltages smaller or equal to 10 mV [8,9,33,39,89]. At higher voltages, they switched into substates. The analysis of their voltage dependence using the Boltzmann formalism showed that the number of gating charges for almost all pores formed by these mitochondrial porins was around two, which means that an e-fold change in *N_o_/N_c_* occurred when the voltage across the membrane was changed by about 12 mV [33,35,39]. The midpoint potential for the distribution of the open and closed pores (i.e., *N_o_ = N_c_*) was in many cases either symmetrical or slightly asymmetrical with values around ±30 mV to ±40 mV [33,35,39,89]. It must be mentioned that the number of gating charges varied somewhat for investigations conducted in different laboratories. Our own results suggested that the number of gating charges was around two, whereas Colombini and coworkers measured about three gating charges in different studies [9,18,64,84].

### 5.3. Selectivity of the Open and Closed Forms of Mitochondrial Pores

The single-channel conductance of hVDAC1 in different salts shown in Table 2 suggests that ions move through the open state of mitochondrial porin in the same way they move in the bulk aqueous phase [33,90]. On the other hand, the pores also exhibit a certain specificity for charged solutes because the single-channel conductance in potassium acetate is somewhat smaller than that in LiCl despite the same aqueous mobility of lithium ions as compared to acetate (see Table 2). This means that the pore formed by hVDAC1 is selective, although it is a wide and water-filled channel in the open state. Experiments with lipid bilayer membranes under zero-current conditions and an externally applied concentration gradient, c″/c′, across the membrane allow for the evaluation of the ionic selectivity of hVDAC1 reconstituted in the membrane. From the asymmetry potential, V_m_, caused by the preferential movement of one sort of ion through the pores, the ratio, P_cation_/P_anion_, of the permeabilities for cations and anions can be calculated using the Goldman–Hodgkin–Katz equation [110].

Table 3 shows the zero-current membrane potentials and the permeability ratios for different mitochondrial porins (porin 31HL (hVDAC), rat liver, yeast, and *Paramecium*) in potassium chloride, potassium acetate, and lithium chloride. It is obvious from the data in Table 3 that the ion selectivity of the mitochondrial porins is dependent on the combination of different anions and cations. The porins are slightly anion-selective (ratio P_anion_/P_cation_ = 1.4–1.7) for potassium and chloride, which have approximately the same aqueous mobility. For the combination of chloride with the less mobile lithium ion (because of its larger hydration shell [110]), the anion selectivity increases. Since the ions move within the channel in a similar way as in the bulk aqueous phase, the channel becomes cation-selective for potassium acetate because of the smaller mobility of acetate compared with that of potassium ions. This result represents another support for the statement of the mitochondrial porin as a wide water-filled pore in the “open” state.

The open state of all mitochondrial porins characterized to date is slightly anion-selective for salts composed of equally mobile cations and anions such as KCl (see above). Mitochondrial porins switch to closed states when the transmembrane voltage exceeds 15–20 mV. The substates have a reduced ion permeability, as the single channel conductance at higher voltages (see Figure 3) and the bell-shaped curves in Figure 5 clearly indicate. The ion selectivity of the closed states cannot be measured using zero-current membrane potential measurements with pores in the closed state because an external voltage must be applied to close the pores. However, the results in Figure 5 and Table 2 suggest that the closed states are cation-selective since, for the combination K-MES (a mobile cation combined with a less mobile anion), the conductance in the open and closed states differs only by a little. The difference between the open state and the closed state of hVDAC1 is more substantial for Tris-HCl (a mobile anion combined with a less mobile cation). This result suggests indeed that the channel is cation-selective in the closed state, although the precise value of the ratio P_cation_/P_anion_ cannot be calculated for the closed states using electrophysiological data.

### 5.4. Single-Channel Conductance of the Closed State of Mitochondrial Porins

The closed state or substate of eukaryotic porins has a reduced permeability for ions, as Figure 3 and Figure 5 clearly indicate. Single-channel conductance experiments allow for an evaluation of the conductance of the closed state at membrane potentials higher than 15–20 mV. At these voltages, the open state of the channels has only a limited lifetime because of its voltage dependence. It is possible to evaluate the single-channel conductance of the closed state by subtracting the conductance of closing events from those of the open state at a voltage of 30 mV. Table 4 shows the results of this type of measurement obtained for three different salts and two types of porins: yeast [37] and VDAC1 from human cells (Porin 31HL [90]). The single-channel conductance of the closed state of the pore was considerably smaller for Tris-HCl than for K-MES, despite a similar aqueous mobility of K^+^ and Cl^−^. This result represents another proof that the closed state(s) of mitochondrial porins is cation-selective.

## 6. Inhibition of the Mitochondrial Pore by a Synthetic Polyanion In Vitro

An amphiphilic, synthetic polyanion (a copolymer with a 10 kDa molecular mass of methacrylate, maleate, and styrene in a 1:2:3 proportion) inhibits dependent on its concentration, different carriers in the mitochondrial inner membrane and the ATPase [111,112]. It is now clear that the effects of the polyanion on mitochondrial metabolism have nothing to do with a direct interaction between the polyanion and inner membrane carriers because such high-molecular-mass molecules cannot penetrate the mitochondrial outer membrane through the porin pores. Moreover, it seems that the polyanion binds to mitochondrial porin and shifts its voltage-dependence in a defined way, thus closing the channel when the membrane voltage has a negative sign at the cis side, i.e., the side where the polyanion is added [92,113,114,115]. A membrane voltage of −10 mV at the cis side was already sufficient to completely switch the pore in the closed configuration when the polyanion was added in a concentration of 100 ng/mL to the cis side of the membrane. For positive potential at the cis side, the channel was always in its open configuration even for voltages up to 120 mV and higher. The effect of the polyanion on hVDAC1 (porin 31HL) is shown in Figure 7. First, the voltage dependence of reconstituted hVDAC1 was studied, which showed the typical bell-shaped curve as already shown in Figure 5. The polyanion was added at a concentration of 100 ng/mL to the cis side. When the voltage had a negative sign on the cis side, the membrane current started to decrease at much smaller voltages than without the polyanion. A membrane potential of −10 mV was sufficient to close the channels almost completely. Higher voltages (see Figure 7) resulted in a complete closure. Positive voltages had no effect on Porin 31HL-induced membrane conductance. This means that the polyanion stabilized the pore in the open state when the sign of the transmembrane potential was positive on the cis side. The addition of the polyanion in the same concentration on both sides resulted in a symmetrical curve with respect to the G/G_0_ axis because the channels closed at small positive and negative potentials. It is noteworthy, that the parameters of pore closure at negative voltages with respect to the addition of the polyanion to the cis side were completely different in terms of Equation (2) compared with those without the polyanion, i.e., n was much larger and reached values of about 4 to 5, instead of about 2, and $V_{0}$ was smaller than 10 mV. The experimental data of different studies suggest that a sidedness for the interaction between the polyanion and different eukaryotic porins does not exist, which means that the polyanion can interact with the gate from both sides of the channel [92,113,114,115].

The use of the polyanion allowed for easy access to the single-channel conductance of the closed state of eukaryotic porins when the polyanion was added in a concentration of 100 ng/mL to the side of the membrane with negative polarity (the cis side). Under these conditions, the single-channel conductance in KCl was about half that of the open state measured at low voltage, which means that the closed channels were still permeable for ions, which makes a direct polyanion-induced block of the channel rather unlikely [93,113,114,115]. It is noteworthy that the polyanion-induced closed state of eukaryotic porins is dependent on the permeability of the single ions through the pore. The polyanion-induced decrease in the single-channel conductance is especially large when a mobile anion (for example, chloride) is combined with a less mobile cation (for example, Tris^+^). In this case, the single-channel conductance was reduced to less than 10% of the open state value, as shown in Figure 5. The effect of the polyanion on combinations of mobile cations with less mobile anions (for instance, on K-MES) was very small (see also Figure 5). This means that the slightly anion-selective eukaryotic pores in the open state highly became cation-selective in the closed state. This was also the result of selectivity measurements in the presence of the polyanion. The ratio P_cation_/P_anion_ for rat liver porin in selectivity measurements without polyanion is approximately 0.6 for KCl (Table 3). For a polyanion concentration of 15 µg/mL on both sides of the membrane, P_cation_/P_anion_ increases to about 10, which suggests that chloride has a very small permeability through the polyanion-mediated closed state of rat liver porin [90]. On the other hand, it is quite clear that positively charged compounds such as calcium ions and other cations have a higher permeability through the closed states of mitochondrial porin/VDAC than through the open state [116].

### The Inhibition of Pore Function by a Polyanion Permits Insight into the Role of Eukaryotic Porins in Mitochondrial Metabolism

The interaction between a polyanion and eukaryotic porins was also studied in intact mitochondria. It inhibited the transport of adenine nucleotides through mitochondrial porins and completely blocked the adenylate kinase located between both mitochondrial membranes [113,114,117]. This means that the experiments with polyanions provided interesting insights into the role of the outer membrane pore in mitochondrial metabolism and the compartmentation of the intermembrane space between the inner and outer membranes [114,117]. The addition of 30 µg polyanion per mg mitochondria completely blocked adenylate and creatine kinases in the intermembrane space. Disruption of the mitochondrial outer membrane by detergent restored full activity of all peripheral kinases, which clearly indicated that compartment formation exists in the intermembrane space of intact mitochondria [117,118]. Similarly, peripheral kinases bound to the pore, such as hexokinase and glycerol kinase were also completely inhibited when mitochondrial but not cellular ATP was utilized, which also indicated that nucleotides must pass the pore to move across the mitochondrial outer membrane [114,117,119]. These results suggest that the mitochondrial porin could be involved in the control of mitochondrial metabolism via its voltage dependence [116,119,120,121]. An important aspect could be the close apposition of mitochondrial inner and outer membranes, which could support the idea that a voltage across the outer membrane is induced via capacitive coupling of inner and outer membranes, in which the folding of the inner membrane may also be involved [8,122]. A polyanion can modify the gate properties of the mitochondrial porin, as it was shown in lipid bilayer experiments. This leads to the exclusion of negatively charged solutes including nucleotides from the pore, which becomes impermeable to them. The closure of mitochondrial porins may be important in the regulation of peripheral kinases like creatine kinase, nucleoside diphosphate kinase, and adenylate kinase, which are located behind the mitochondrial outer membrane [113]. The experiments with polyanions provided interesting insight into the role of eukaryotic porins in mitochondrial metabolism, as described in this section. However, their use as a therapeutic molecule is definitely not possible because of their high anionic charge and high molecular mass, which do not allow for polyanions to cross the cytoplasmic membranes of cells. The search for alternative molecules that could modulate the permeability properties of eukaryotic porins is discussed in some detail in Section 8 of this article.

## 7. Structure of the Mitochondrial Outer Membrane Pore

X-ray crystallography of mitochondrial porins was not possible for many years despite many attempts. This means that their folding in tertiary structure was a matter of debate between different groups. Colombini and coworkers favored a pore containing 12 to 13 β-strands in combination with the N-terminal α-helix as part of the channel wall [123,124,125,126,127]. Our own approach favored many antiparallel and amphipathic β-strands tilted at an angle with respect to the surface of the outer membrane, like the situation in bacterial porins, which also agreed with models of the mitochondrial outer membrane pore [33,128]. The position of the amphipathic α-helical structure represented a major problem for the validity of the previous models because it should somehow be involved in the stability of the pore and the gating mechanism, as experiments with N-terminal deletion porins suggested [68,74,129,130]. However, some information was possible from the study of two-dimensional crystals of fungal mitochondrial outer membranes [131,132,133,134]. According to the Fourier-filtered electron microscopic images of the crystalline mitochondrial outer membrane arrays, the pore appears as a cylinder normal to the membrane plane with an outer diameter of about 3.8 nm for the polypeptide backbone and an inner diameter of about 2.5 nm [132,133].

The rough structure of eukaryotic porins as obtained by electron microscopic images was verified by three groups, which successfully derived the 3D structure of eukaryotic porins simultaneously at high resolution using different techniques [135,136,137]. Hiller et al. [136] used the technique of solution NMR to study recombinant hVDAC1 reconstituted in detergent micelles. In this case, the location of the N-terminus was not resolved in the images. Bayrhuber et al. [135] derived the 3D structure of hVDAC1 from a combination of NMR spectroscopy and X-ray crystallography. Ujwal et al. [137] succeeded in crystallizing murine VDAC1 (mVDAC1) to resolve its 3D structure. The three images agreed on the basic structure of the mitochondrial pore that is formed by a β-barrel cylinder with 19 β-strands [135,136,137]. Eighteen β-strands (1 to 18) are pairwise antiparallel, similar to the situation in bacterial porins. Β-strands nineteen and one are in a parallel configuration. Two 3D structures show the location of the N-terminal α-helix horizontally midway in the pore, restricting its size [135,137]. This means that the α-helix has a strategic position to control the passage of metabolites and ions through the mitochondrial pore, as shown in the schematic picture of the 3D structure of murine VDAC1 in Figure 8, despite some criticism based on the assumption that the published structure differs from the functional structure [64].

The comparison of the two 3D structures shown in Figure 8 demonstrates that the architecture of the two outer membrane pores of mitochondria and Gram-negative bacteria is quite similar. This presumably is related to the history of bacterial and mitochondrial outer membrane pores as it was discussed in the introduction of this article. It is also noteworthy that the translation and assembly of both pores are very similar and are driven by β-strands [139,140,141,142,143,144]. The β-strands of both β-barrel cylinders are tilted by 30° to 40° toward the surface of the membranes. The dimensions of the eukaryotic porin are 35 Ă for the height and 40 Ă for the width. The N-terminal α-helix (amino acids 1 to 21) is located inside the β-barrel cylinder and acts as a gate, but it is also a stabilizing element for the mitochondrial pore similar to external loop 3 of OmpF that is folded inside the bacterial pore [137,138]. Despite the location of the N-terminus inside the eukaryotic pore, it has a high ion permeability, which means that the conductance of mVDAC1 is approximately the same as that of OmpF trimers, i.e., it is considerably higher than that of the OmpF monomer, which has a single-channel conductance of 1.4 nS in 1 M KCl [7,39]. Tom40, which represents the major component of the mitochondrial outer membrane import machinery, is also a member of the VDAC family and shows the same structure of 18 antiparallel β-strands and one pair of parallel β-strands [73,145,146]. The most interesting point in the comparison of bacterial and mitochondrial porins is the fact that bacterial outer membrane pores have only passive properties, whereas, during evolution, mitochondrial porins adopted an active role in mitochondrial metabolism, which is discussed in some detail here.

### 7.1. Functional Amino Acids in the 3D Structure of Mitochondrial Porins

When pig heart mitochondria are treated with low doses (1.5 nmol/mg of mitochondrial protein) of C14-labeled dicyclohexylcarbodiimide (DCCD), three mitochondrial polypeptides of approximately 9, 16, and 33 kDa bound DCCD [147,148]. The two smaller DCCD-binding proteins are parts of the F_0_F_1_-ATPase localized in the mitochondrial inner membrane [147]. The 33 kDa DCCD-binding protein present in the outer membrane of pig heart mitochondria was identified as the eukaryotic porin based on biochemical evidence and electrophysiological experiments, although DCCD-binding bovine heart mitochondrial porin did not change the electrophysiology of the pore [148]. However, labeling porin with DCCD resulted in the loss of hexokinase binding to porin [149,150] because the porin was identified as the hexokinase-binding protein [19,151]. Fifty percent inhibition of hexokinase binding occurred at very low concentrations of DCCD of less than 2 nmole of DCCD/mg of mitochondrial protein [150]. Water-soluble carbodiimides had no effect on hexokinase binding on porin, indicating that the binding place was in a hydrophobic environment. DCCD binding to proteins suggested that a negatively charged amino acid exists in a hydrophobic environment [148,150]. This amino acid was identified as glutamate 72 in the sequence of bovine heart eukaryotic porin [152]. The role of this negative charge in the mitochondrial metabolism of the three VDAC isoforms in Zebrafish was studied in detail recently because homologous glutamate 73 is present in VDAC1 (E73; see Figure 9) and VDAC2 but not in VDAC3 and plays an important role in the regulation of Ca^2+^ uptake in mitochondria [153]. Mutations of E73 did not change the electrophysiology of mVDAC similar to the case of DCCD binding to E72 described above for bovine heart porin [154].

Alignments of the primary sequences of eukaryotic porins show that a high diversity exists between the single sequences. When porin sequences of Porin 31HL, *Paramecium*, yeast, potato, *Neurospora crassa*, and *Drosophila* were aligned, only 15 amino acids were identical within approximately 280 amino acids in total [39]. This demonstrates the high possibility of variability in amino acids in β-strands. Interestingly, a triplet of amino acids exists, which are preserved in many primary sequences around 90 to 100 amino acids (see Figure 9). This triplet consists of mVDAC1 in glycine G94, leucine L95, and lysine K96. The high conservation of this triplet in many eukaryotic porins suggests in principle an important role of the three amino acids in mitochondrial porin function [39,129,145]. However, the role of the conserved triplet GLK is still unknown because it is not involved in pore formation or binding of nucleotides, but it contributes to anion selectivity because the GLE mutant of mitochondrial porin of *Neurospora crassa* is cation-selective, similar to the GLE mutant of yeast porin [124,129].

In addition to the above-discussed preserved amino acids in the primary sequences of Porin 31HL, *Paramecium*, yeast, potato, *Neurospora crassa*, and *Drosophila melanogaster*, several other amino acids are also preserved [39]. These are D15, K19, Y21, L81, T83, P135, G152, N215, D228, D263, K274, and G276 (Porin 31HL numbering). The role of these preserved amino acids, particularly those of positive and negative charge in the structure and function of the corresponding mitochondrial porins, needs to be elucidated in the future. This may also apply to the function of cysteines within the primary sequence of human mitochondrial porin. Porin 31HL (hVDAC1) contains two cysteines, C127 and C232 [43,52]. Bovine heart porin, which contains cysteines in similar positions as Porin 31HL, was investigated for the role of the two cysteines in porin function [43]. Reduced forms of the porin show the same pore-forming characteristics as the oxidized forms when they are reconstituted into artificial lipid bilayer membranes [43]. C127 is localized within β-strand 8 and adopts a more hydrophobic position within this β-strand [43,137]. The other cysteine (C232) is more hydrophilic according to its location in β-strand 16 [137]. Both cysteines reacted to labeling by eosin-5-maleimide and N-(1-pyrenyl)-maleimide in a study of bovine heart porin [43]. This reactivity suggested in principle that the two cysteines are localized in a region of relatively high dielectric constant within the membrane, which is in between the lipid and water phases [43]. The labeling of both cysteines with eosin-5-maleimide and N-(1-pyrenyl)-maleimide could be inhibited by N-ethylmaleimide, which suggested that both cysteines are sensitive to N-ethylmaleimide [43]. However, C127 appeared to be more sensitive to N-ethylmaleimide, which is presumably caused by its location on the more hydrophobic side of the amphipathic β-strand 8 of mitochondrial porins [43].

### 7.2. The N-Termini of Mitochondrial Porins Are Responsible for Voltage-Dependent Gating

The important part of mitochondrial porins that is involved in voltage-dependent gating is the N-terminal α-helix, which is localized about halfway within the β-barrel cylinder (see Figure 8 and Figure 9). The deletion of amino acids 1–20 from the N-terminal end of *Neurospora crassa* porin and hVDAC1, and the deletion of amino acids 1–31 of hVDAC2 completely abolished the voltage-dependent gating of the pores formed by these eukaryotic porins [68,74,96]. Simultaneously, the average single-channel conductance of the deletion mutants decreased and assumed values near those of the voltage-driven closed pores. The pores formed by the deletion mutants in lipid bilayers at small voltages were not stable with time and switched frequently among conductance levels below those formed by the native mitochondrial porins [68,74,96]. These results indicated that the gate within the eukaryotic pores, i.e., the α-helix formed by the N-terminal amino acids 1–20 is not only responsible for gating but functions also as a stabilizing element for the pore structure. Its removal or its mutation may lead to substantial changes within the β-barrel structure of the mitochondrial pore [74,96,155,156,157]. Regarding the mechanism of gating, it has been proposed that major structural rearrangements of the α-helix and the β-barrel cylinder are responsible for gating [74,125,155,156,157]. However, a mVDAC mutant, where a disulfide bond could be formed between neighboring amino acids L10C (within the α-helix) and A170C (located in β-strand 11), showed almost normal voltage-dependence as compared with a wild-type porin [158]. This result makes substantial changes in the structure of the N-terminal α-helix, such as pulling it out from the lumen of the pore during gating rather unlikely, because the α-helix remains associated with β-strand 11 during voltage gating in the mVDAC mutant L10C coupled with A170C with a disulfide bond [157,158].

Figure 9 shows also amino acids that are involved in the voltage gating of mVDAC1 and related mitochondrial porins of mammals. Mutations of lysine 12 (corresponding to arginine in some primary sequences [39]) demonstrated that this amino acid has a high impact on the channel gating of mVDAC1 [159]. The mutation K12E, i.e., the exchange of lysine 12 with glutamate completely abolished the voltage-dependence of mVDAC. This means presumably that the residue K12 has a high influence on β-barrel fluctuations and force-gating transitions of mitochondrial porins [159]. In addition to K12, leucine 10 also has an important influence on the voltage-dependent gating of mitochondrial porins. The mutation of L10N within hVDAC led to a substantial change in the conductance fluctuations as compared with the conductance steps of wild-type hVDAC [74]. L10 is localized within a hydrophobic environment created by valine 143 (within β-strand 9) and alanine 170 (within β-strand 11). Its replacement with the highly hydrophilic amino acid asparagine leads to some distortion of the β-barrel cylinder [74]. This also influences the voltage dependence of the L10N mutant, which is reduced as compared with that of the wild-type hVDAC1 [74].

## 8. Search for Modulators of the Function of Eukaryotic Porins—Correlation between Eukaryotic Porins and Cancer and Apoptosis

Mitochondrial porins control the flux of many metabolites between mitochondria and the cytosol. In their open state, they allow for the transport of anionic solutes, in particular, substrates like glycolytic ATP, phosphate, and small cations, into mitochondria. Simultaneously, mitochondrial ATP and other metabolic molecules move to the cytosol. The voltage-induced closed states of mitochondrial porins [113,115] favor the transport of cations, in particular, Ca^2+^ and other cations, and inhibit the transport of anionic solutes through the pore. Cancer cells are very often characterized by boosted aerobic glycolysis, also known as the so-called Warburg phenotype, which is also accompanied by suppressed mitochondrial metabolism [160,161]. However, mitochondrial metabolism is quite flexible, which means that cancer cells can swap between predominantly glycolytic or oxidative phenotypes. The states of mitochondrial porins play an important role in these processes. The closed states favor aerobic glycolysis, whereas the open state of mitochondrial porins promotes oxidative phosphorylation and reduces glycolysis. Glycolysis is a low-energy process, which provides only two moles of ATP per mole glucose. The yield of oxidative phosphorylation is more than 30 moles of ATP per mole glucose, which means that it is much higher compared with glycolysis. Mitochondrial porins are considered governors or gatekeepers of mitochondrial metabolism in these processes, which are supported by the effects of porin deletions [24,121,162,163,164,165,166,167,168,169].

This means that the transport of substrates through the mitochondrial outer membrane pore has also an important impact on the physiology and apoptosis of cancer cells. In recent years, several research groups have searched for molecules that could interact with hVDAC1 as blockers or openers to influence mitochondrial metabolism [80,165,170,171,172]. Steroids and hydrophobic drugs, such as olesoxime, efsevine, and propofol, can also change the gating behavior of hVDAC1. They can modulate mitochondrial metabolism by direct interaction with hVDAC1 causing a change in its permeability properties [172]. Several peptides (mastoparan, mitoparan) interact also with eukaryotic porins and modulate their permeability properties [170]. Eukaryotic cells can contain up to six members of the tubulin superfamily. Two of them, α- and β-tubulin bind to hVDAC1 and change its permeability for ionic solutes, probably by the interaction between the disordered polyanionic C-terminal domain of tubulins and the gate within hVDAC1 [172,173]. This means that tubulin and other molecules with polyanionic C-terminus domains interact with mitochondrial porins in a way that is similar to the interaction between polyanion and porin, which means that voltage dependence is drastically enhanced in the presence of these molecules and may lead to the inhibition of mitochondrial metabolism in cancer cells [172]. Similarly, G3139, a phosphorothioate oligonucleotide, is also a VDAC modulator that presumably blocks the pore for the passage of mitochondrial metabolites by an interaction with the gate in VDAC [157,174]. These examples demonstrate that a variety of molecules can interact with eukaryotic porins and modify their characteristics either by interacting with the gate and/or binding inside the pore. Of special interest are compounds that open the closed form of eukaryotic porins because they decrease glycolysis and increase the cytosolic ATP/ADP ratio driven by oxidative phosphorylation. The effects of modulators on eukaryotic porins were described in full detail in a recent review by Heslop et al. [165]. This also includes molecules that interact with the pore and act as pore blockers similar to polyanion or as pore openers in cancer cells. The effect of these molecules on VDAC1 was described in detail by De Pinto and coworkers [166]. Further study of these effectors may lead to an important step forward in the treatment of diseases that are combined with the permeability properties of the mitochondrial outer membrane, i.e., of eukaryotic porins [175].

Intrinsic apoptosis of eukaryotic cells may be triggered when proapoptotic molecules, like cytochrome C, leak out of the mitochondria. The basic mechanism of the permeability of cytochrome C through the mitochondrial outer membrane after its release from cardiolipin at the inner mitochondrial membrane is still a matter of debate. It could occur through pores in the outer mitochondrial membrane that are formed from dimeric Bak and Bax [176,177]. Another possibility is passage through the proapoptotic VDAC1 pore. However, cytochrome C has a molecular mass of about 12 kDa, which means that it is by far too big to diffuse through the VDAC1 pore with an exclusion limit of about 5 kDa ([33]; see also Section 4 of this article). To overcome this problem, it has been suggested that because of calcium-dependent overproduction, VDAC1 aggregates within the outer mitochondrial membrane and forms larger pores in the aggregated state, allowing for the passage of cytochrome C [24,178]. These giant outer membrane pores have not been observed in reconstitution experiments with lipid bilayer membranes. On the other hand, it is also possible that the aggregation of VDAC1 leads to local instability of the mitochondrial outer membrane, thus allowing the leakage of compounds from the intermembrane from the defects in the cytosol. A few leaks or giant pores would be sufficient for the passage of cytochrome C and the onset of apoptosis.

## 9. Conclusions

Eukaryotic porins deeply embedded in the mitochondrial outer membrane of eukaryotic cells are localized in an important strategic position between the cytosol and mitochondria. In the open state, they have an extremely high permeability for anionic mitochondrial solutes. Peripheral kinases bind to the pores and utilize directly the ATP generated by oxidative phosphorylation. The pores are lined up by 19 amphipathic β-strands and form a β-barrel cylinder similar to their ancestors, the bacterial porins. However, in contrast to them, they have also active sieving properties, which allows them to act as a governor or gatekeeper of mitochondrial metabolism. The gate is given by the N-terminal α-helix, about 20 amino acids long, localized approximately in the middle of the β-barrel cylinder in the open configuration, where mitochondria are in the oxidative phenotype (anti-Warburg) Driven by small transmembrane voltages, the eukaryotic porins switch in ion-permeable substates that have remarkably different permeability properties than the open state. Calcium ions and other cations have a high permeability through the closed states, but anionic solutes are widely excluded from the pores, which presumably means that oxidative phosphorylation is inhibited under these conditions. The exact mechanism of voltage-dependent gating is not fully understood, but it seems that charged and neutral amino acids at the gate interact with components of the β-barrel cylinder. The closed states of eukaryotic porins favor aerobic glycolysis (pro-Warburg) in the cells, which means that mitochondrial metabolism is suppressed under these conditions. Drugs that favor the opening of mitochondrial porins can initiate the death of cancer cells, where VDAC1 is an important pro-apoptotic factor, either directly or through other pore-formers. Further studies of compounds that interact with mitochondrial porins for opening or closing are of great interest because these investigations may allow for the development of new therapies against cancer cells.

## Figures and Tables

**Figure 1 biomolecules-14-00303-f001:**
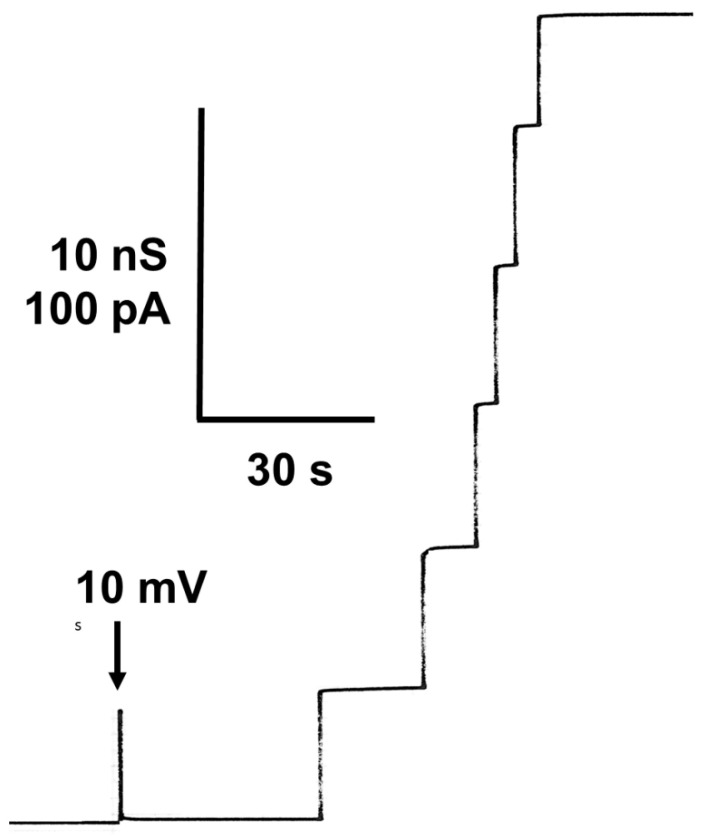
Stepwise increase in the membrane current (given in pA) after the addition of hVDAC1 (also known as porin 31HL) to a black lipid bilayer membrane given as a function of time. The aqueous phase contained 5 ng/mL porin 31HL and 1 M KCl [90]. The membrane was formed from diphytanoyl phosphatidylcholine/n-decane. The applied voltage was 10 mV and T = 20 °C. The arrow indicates the onset of the voltage (10 mV).

**Figure 2 biomolecules-14-00303-f002:**
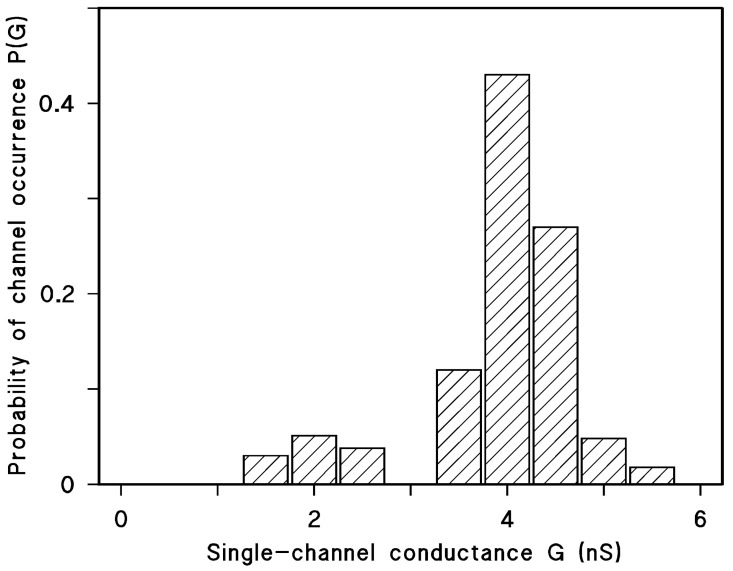
Histogram of conductance fluctuations like those shown in Figure 1, which were observed with membranes of diphytanoyl phosphatidylcholine/n-decane in the presence of Porin 31HL [90]. P(G) is the probability for the occurrence of a conductance step with a certain single-channel conductance (given in nS). The aqueous phase contained 1 M KCl. The voltage applied was 10 mV. The mean value of all upward directed steps was 4.3 nS for the right-side maximum and 2.3 nS for the left-side maximum (in total 176 single events); T = 20 °C.

**Figure 3 biomolecules-14-00303-f003:**
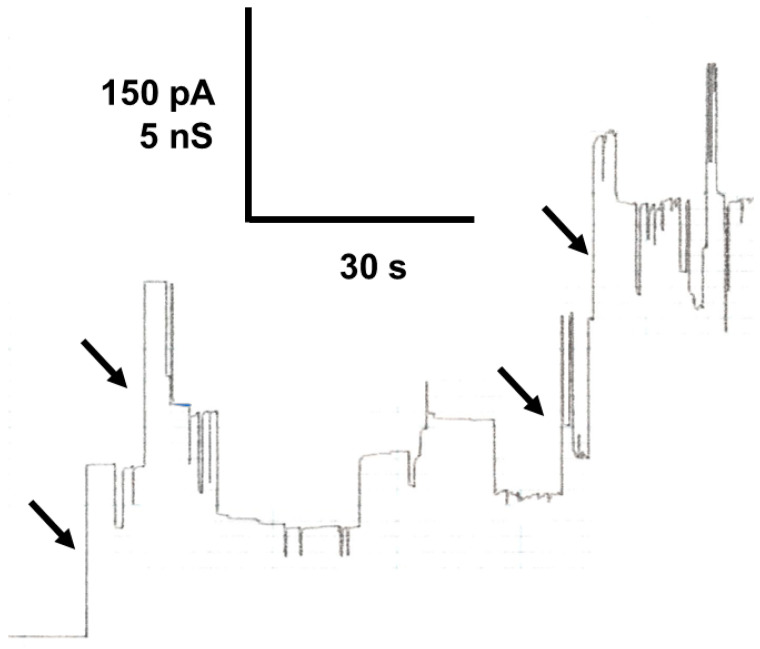
Voltage-dependence of Porin 31HL (hVDAC1). A voltage of 30 mV was applied to a membrane formed from diphytanoyl phosphatidylcholine/n-decane in 1 M KCl. Then, Porin 31HL was added to one side of the membrane at a concentration of 5 ng/mL under stirring. The reconstitution of Porin 31HL in the membrane occurs in large current steps of about 120 pA (4 nS; arrows). The voltage dependence of Porin 31HL results in subsequent decreases in the current in substates that are not stable and show on- and off-kinetic behavior. T = 20 °C.

**Figure 4 biomolecules-14-00303-f004:**
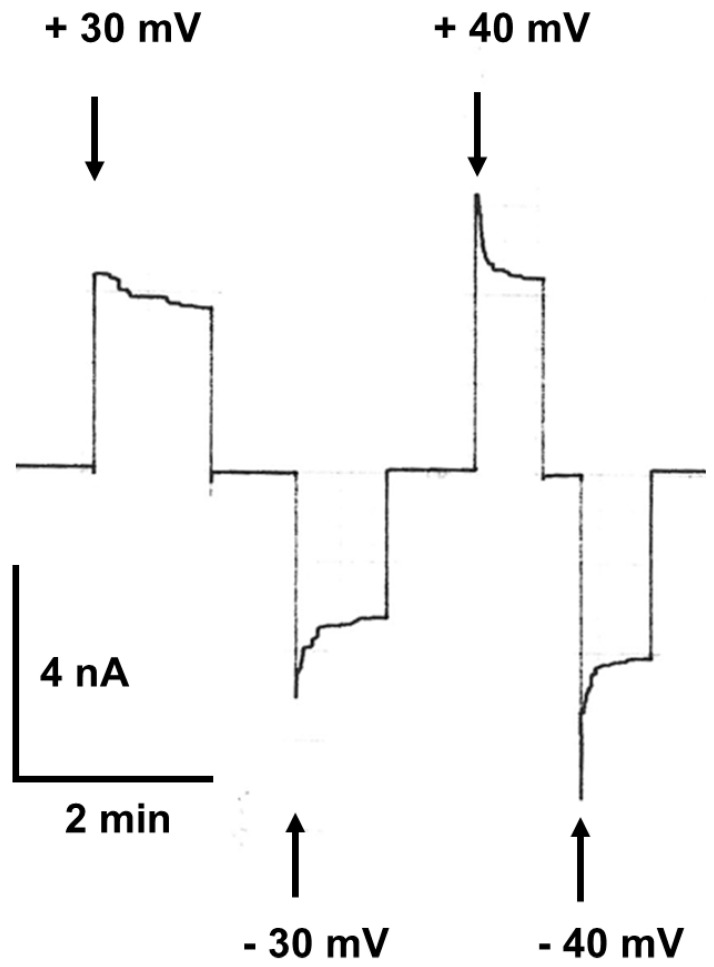
The voltage dependence of Porin 31HL measured in a multichannel experiment. About 50 pores were reconstituted in a membrane from diphytanoyl phosphatidylcholine/n-decane. The voltage across the membrane was switched to 30 mV (with respect to the cis side, the side of the addition of 5 ng/mL protein) and then to −30 mV, followed by 40 mV and −40 mV. The channels switched to substates of the open state in a single exponential curve. The aqueous phase contained 0.5 M KCl, T = 20 °C.

**Figure 5 biomolecules-14-00303-f005:**
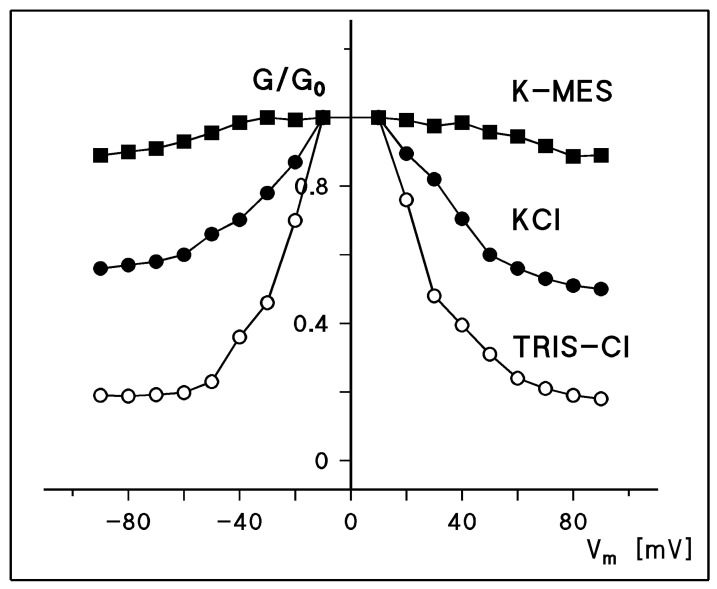
The ratio of the conductance, G, at a given voltage, V_m_, divided by the conductance, G_0_, at 10 mV as a function of the voltage. The aqueous phase contained either 0.5 M KCI, 0.5 M K-MES, or 0.5 M TRIS-HCI (pH in all cases 7.2). The cis side contained about 10 ng/mL hVDAC1 (Porin 31 HL [90]). The sign of the voltage is given with respect to the trans side, the side opposite to the addition of Porin 31HL.

**Figure 6 biomolecules-14-00303-f006:**
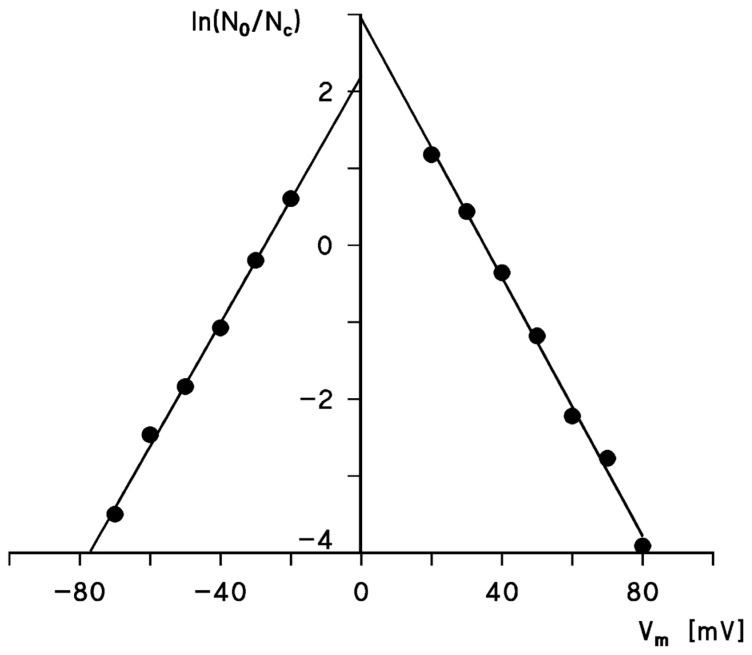
Semilogarithmic plot of the ratio, *N_o_/N_c_*, as a function of the transmembrane potential V_m_. The data were taken from Figure 5. The slope of the straight lines is such that an e-fold change in *N_o_/N_c_* is produced by a change in V_m_ of 12.5 mV (left side) and 11.9 mV (right side), corresponding to gating charges *n* = 2.0 and 2.1, respectively. The midpoint potential of the *N_o_/N_c_* distribution (i.e., *N_o_ = N_c_*) was at 27.4 mV (left side) and 35 mV (right side).

**Figure 7 biomolecules-14-00303-f007:**
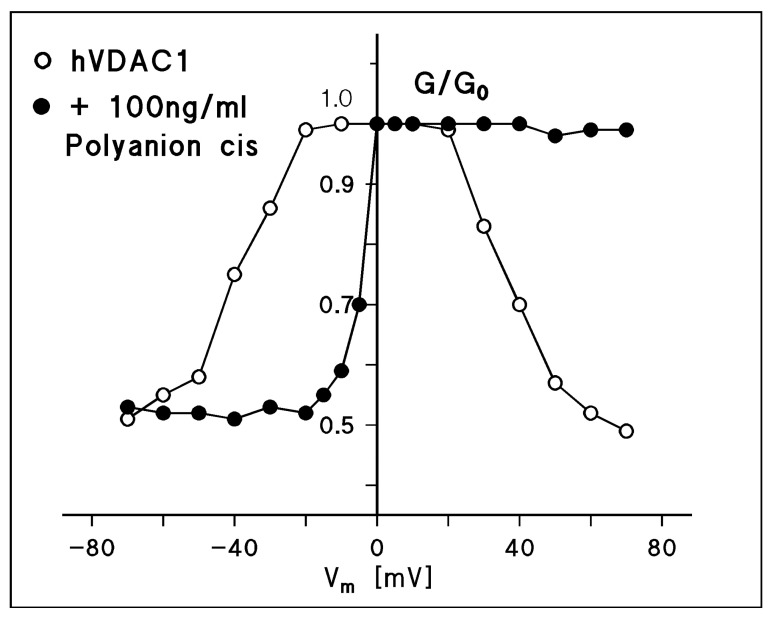
Voltage dependence of hVDAC1 without and with polyanion. The open circles show the control. Different voltages were applied to pores formed by hVDAC (Porin 31HL) in a lipid bilayer of diphytanoyl phosphatidylcholine/n-decane, and G/G_0_ was calculated from the decay in the conductance. Then, 100 ng/mL of polyanion was added to the cis side of the membrane, and the voltage dependence was measured again (closed circles). Note that the voltage dependence changed completely in the presence of the polyanion; 1 M KCl; T = 20 °C.

**Figure 8 biomolecules-14-00303-f008:**
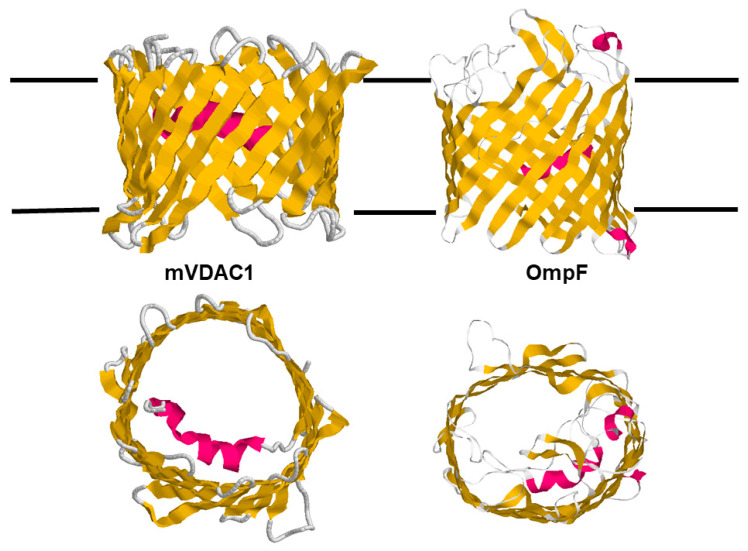
Three-dimensional structures of the mitochondrial outer membrane pore (mVDAC1) and an OmpF monomer of the *Escherichia coli* outer membrane. β-strands within protein structures are shown in yellow and α-helical stretches are shown in red. The 3D structure of mVDAC1 is shown from the side with the N- and C-termini of the protein up (probably directed to the surface of the mitochondrion). The side of the bacterial porin (upper structure) is shown with the surface of the bacterial cell up. The view from the top of mVDAC1 is shown from the N- and C-termini of mVDAC1. The view of OmpF is shown from the surface of the bacterial cell (structures down). mVDAC1 (PDB code: 3emn.pdb) is the 3D structure of mouse mitochondrial porin as obtained by ref. [137] by crystallization of murine VDAC1. OmpF (PDB code: 2OMF) represents the 3D structure of the major outer membrane protein of *E. coli* [138].

**Figure 9 biomolecules-14-00303-f009:**
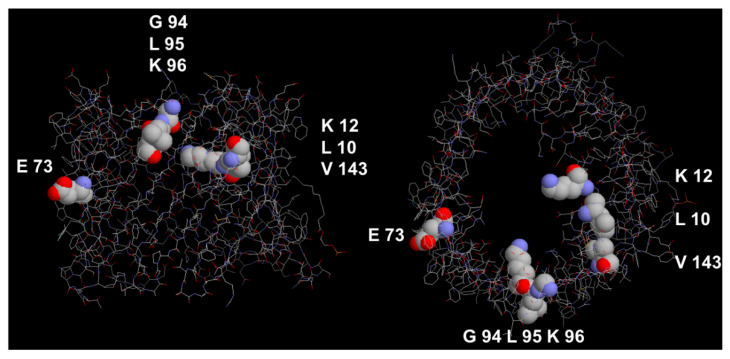
Location of important amino acids within the 3D structure of murine mitochondrial porin 1 (mVDAC1) (PDB code: 3emn.pdb). The left panel shows a side view of the structure with the N- and C-termini up. The right panel shows the structure from the N- and C-termini down perpendicular to the membrane plane. The important amino acids are indicated in the one-letter code and shown in spacefill. PDB data were taken from ref. [137].

**Table 1 biomolecules-14-00303-t001:** Single-channel conductance of mitochondrial (eukaryotic) porins (VDACs) from different eukaryotic organisms.

Mitochondrial Porin (VDAC)	G (nS)	References
Human VDAC1 (Porin 31HL)	4.3 4.1	[90] [54]
Human VDAC2	4.0 2.0 and 4.0	[54] [96]
Human VDAC3	3.9	[97]
Rat liver	4.3	[16]
Beef heart	4.0	[98]
Rabbit liver	4.0	[98]
Rat brain	4.0	[35]
Rat kidney	4.0	[35]
Pig heart	3.5	[35]
*Anguilla anguilla*	4.0	[99]
*Drosophila melanogaster* VDAC CG17140	4.5 3.4/1 M NaCl	[92] [100]
*Protophormia*	4.5	[101]
*Neurospora crassa*	4.5	[15]
Yeast	4.5 4.2 4.2	[51] [37] [91]
*Paramecium*	4.5 2.4	[10] [102]
*Pea* mitochondria	1.5 and 3.7	[94]
*Pea* root plastids	1.5 and 3.7	[46]
*Maize* root plastids	1.5 and 3.7	[46]
*Solanum tuberosum* POM 34	2.0 and 3.5	[45]
*Maize* mitochondria	1.5 and 3.7	[38]
*Phaseolus coccineus*	3.7	[93]
*Arabidopsis*	0.5/300 mM KCl	[103]

**Table 2 biomolecules-14-00303-t002:** Average single-channel conductance of Porin 31HL (hVDAC1) in different salt solutions.

Salt	c (M)	G (nS)
KCl	0.01	0.05
	0.03	0.15
	0.1	0.45
	0.3	1.3
	1.0	4.3
	3.0	11
LiCl	1	3.2
K-acetate	1	1.5
Tris-Cl	0.5	1.5
Tris-HEPES	0.5	0.18
K-MES	0.5	0.70

**Table 3 biomolecules-14-00303-t003:** Zero-current membrane potentials, V_m_, of membranes from diphytanoyl phosphatidylcholine/n-decane in the presence of rat liver [16], yeast [37], and *Paramecium* [102] porins measured for a 10-fold gradient of different salts *.

Salt	V_m_ (mV)	P_anion_/P_cation_
Rat liver		
KCl (pH 6)	−11	1.7
LiCl (pH 6)	−24	3.4
Potassium acetate (pH 7)	+14	0.50
Yeast		
KCl (pH 6)	−7	1.4
LiCl (pH 6)	−20	2.6
Potassium acetate (pH 7)	+14	0.5
*Paramecium*		
KCl (pH 6)	−11	1.7
LiCl (pH 6)	−24	3.4
Potassium acetate (pH 7)	+14	0.50

* V_m_ is defined as the potential of the dilute side (10 mM) relative to that of the concentrated side (100 mM). P_anion_/P_cation_ was calculated using the Goldman–Hodgkin–Katz equation [110].

**Table 4 biomolecules-14-00303-t004:** Average single-channel conductance of the open and closed states of yeast [37] and Porin 31HL [90] in different 0.5 M salt solutions. The pH of the aqueous salt solutions was adjusted to 7.2. The protein concentration was between 5 and 10 ng/mL; V_m_ = 30 mV and T = 25 °C. The single-channel conductance of the closed state was calculated by subtracting the conductance of the closing events from the conductance of the initial opening of the pores.

Salt	Open State (nS)	Closed State (nS)
Yeast porin		
KCl	2.3	1.3
K-MES	0.95	0.65
Tris-HCl	1.5	0.30
Human porin (Porin 31 HL)		
KCl	2.4	1.4
K-MES	0.70	0.65
Tris-Cl	1.5	0.30

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
