# Peer review of "Solute Transport through Mitochondrial Porins In Vitro and In Vivo"

_biomolecules, 2024, doi:10.3390/biom14030303_

Round 1

Reviewer 1 Report

Comments and Suggestions for Authors

This paper is a comprehensive overview of the mitochondrial channel VDAC with an emphasis on the Author’s impressive series of papers devoted to studies of this channel from the late 1970s to now. This review demonstrates Dr. Benz’s impressive contribution to the field of VDAC research through all those years. The Author provides a detailed excursion into the history of VDAC’s discovery, identification, and systematic biophysical studies of its channel properties. However, from my point of view, the part of the review devoted to the description and comparison of a variety of earlier protocols of VDAC purification and isolation from different sources is disproportionately large and simply not needed. I think that today, when cell-free assays are widely used in different labs for the preparation of individual VDAC isoforms and VDAC proteins are commercially available, that such a detailed and long description does not have scientific relevance and would not be interesting for readers, especially young ones. I have the same concern regarding the long description of the different proposed folding patterns of VDAC in chapter 7. I do not think that such a long historical excursion is relevant now, after the VDAC1 structure has been solved. I also do not understand why the Author persistently keeps calling hVDAC1 “human Porin 31H” or eukaryotic porin, when “VDAC” is a widely accepted and well-recognized name for this protein. I think that this terminology would create unwanted confusion for the wide audience of this Journal. I believe that for many years now, “porin” has been an established name for beta-barrel channels of the bacterial outer membrane. There is no genetic connection between the mitochondrial beta-barrel VDACs and bacterial porins.

The review is well-structured and illustrated but it needs a vigorous editing of English grammar – some sentences are incomprehensible.  There are too many typos to list them. In the existing form, it cannot be published.

I have a few more comments:

1.     I appreciate the fact that the literature about VDAC is enormous and has kept rapidly growing since 2008 when the VDAC1 3D structure was solved, thus making a selection of what to cite and how to focus a review a serious challenge for any author. However, I cannot agree with the Author’s choice to cite almost exclusively his own works in Chapter 5 on the electrophysiology of mitochondrial porins and almost ignore Colombini’s group’s works on VDAC channel biophysics. Here are just a few examples:

-       Page 11: “…presumably because the selectivity of the pore in the open and closed configuration is different.” This is not a “presumable” situation” - the selectivity of open and closed VDAC states has been experimentally measured by Colombini (J Membr Biol 1989) in the KCl gradient. These are published data.

-       Page 15: “The ion selectivity of the closed states cannot be measured using zero-current membrane potential measurements with pores in the closed state because an external voltage must be applied to close the pores.” This is simply incorrect. Please read Colombini’s works on VDAC selectivity and how to measure it. See e.g., Tan & Colombini BBA, 2007.

-       Page 13: “This result suggested again that the number of gating charges involved in channel gating is about two”. This is incorrect: VDAC gating charge was experimentally measured by Colombini and coauthors and by other labs and is ~ 3e. This is based on many different publications on VDAC channel gating.

2.     Chapter 6: “at the cis-side the channel was always in its open configuration even for voltages up to 120 mV and higher”. Does the Author have any explanation for such an unusual VDAC behavior in the presence of the polyanion? The following statement regarding bacterial porins that “the polyanion can interact with the gate from both sides of the channel” seems to contradict the effect of the polyanion on VDAC gating (Fig. 5).

-       It is not clear why the Author thinks that “a direct polyanion-induced block of the channels is rather unlikely”?  What is the experimental evidence for such a conclusion? On page 23 the Author compares, quite rightly, the interaction of polyanion and VDAC with polyanionic C-terminus of tubulin, which is a permeation block.

-       Why is the increase of cation selectivity in the presence of polyanion surprising if the polyanion-induced closed VDAC state is always more cation-selective and selectivity depends on the type of anion and its concentration? This has been described in a few works by different groups. See, e.g., a work on VDAC selectivity (Zambrowicz & Colombini BJ 1993) and OmpF selectivity (Alcaraz et al BJ 2009 doi:10.1016/j.bpj.2008.09.024).

3.     Part 6.1: the way it is written gives the impression that the effect of the polyanion on mitochondrial adenylate and creatine kinases is the only reliable result indicating that ATP crosses the outer membrane through VDAC. Maybe the Author did not mean it, but that is how this paragraph sounds. I thought that the translocation of ATP through the VDAC open state and the lack of translocation through the closed state are well-established facts demonstrated experimentally by Colombini’s group in 1996 (see e.g., doi: 10.1074/jbc.271.45.28006).

4.     Chapter 6.1, page 18: in the discussion about “mitochondrial porin could be involved in the control of mitochondrial metabolism via its voltage-dependence” the theoretical works of Lemeshko on the role of VDAC-hexokinase complex in the generation of the potential across the outer membrane seem worth mentioning in the context of a general discussion of this paper.

5.     Chapter 7: The similarities between VDAC1 and OmpF structures are quite exaggerated in this chapter. The main similarity between them is that they both are both beta-barrel channels. The number of beta-strands is different, the construction of the constriction zone – alpha-helical N-terminus for VDAC and the loop for OmpF– are different, and the conductance of each monomer is different – 4 nS for VDAC and 1.4 nS for OmpF monomer in 1 M KCl. The comparison of monomer VDAC conductance with trimeric OmpF is not adequate.

The references for Tom40 structure should be updated with the latest publications: Tucker and Park, Nature Struct Mol Biol 2019, and Araiso et al Nature 2019.

6.     Could the statement on Page 20 “the fact that bacterial outer membrane pores have only passive properties, whereas mitochondrial porins adopted during evolution an active role in mitochondrial metabolism” be clarified? Bacterial and mitochondrial porins form passive diffusion pores. Their physiological roles are different.

7.     Part 7.1, page 21: I believe that it should be deletion of 1-31 residues in VDAC2 in Gattin et al, 2015.

8.     Chapter 8: The statement “The voltage-induced closed states of mitochondrial porins (Benz et al., 1990) favor the transport of cations, in particular of Ca2+” needs a reference  to Tan&Colombini BBA 2007, who showed that VDAC closed states are significantly more permeable for calcium than the open state.

Comments on the Quality of English Language

The paper needs thorough proofreading for English grammar and typos before it can be published in your Journal. 

Author Response

This paper is a comprehensive overview of the mitochondrial channel VDAC with an emphasis on the Author’s impressive series of papers devoted to studies of this channel from the late 1970s to now. This review demonstrates Dr. Benz’s impressive contribution to the field of VDAC research through all those years. The Author provides a detailed excursion into the history of VDAC’s discovery, identification, and systematic biophysical studies of its channel properties. However, from my point of view, the part of the review devoted to the description and comparison of a variety of earlier protocols of VDAC purification and isolation from different sources is disproportionately large and simply not needed. I think that today, when cell-free assays are widely used in different labs for the preparation of individual VDAC isoforms and VDAC proteins are commercially available, that such a detailed and long description does not have scientific relevance and would not be interesting for readers, especially young ones. I have the same concern regarding the long description of the different proposed folding patterns of VDAC in chapter 7. I do not think that such a long historical excursion is relevant now, after the VDAC1 structure has been solved. I also do not understand why the Author persistently keeps calling hVDAC1 “human Porin 31H” or eukaryotic porin, when “VDAC” is a widely accepted and well-recognized name for this protein. I think that this terminology would create unwanted confusion for the wide audience of this Journal. I believe that for many years now, “porin” has been an established name for beta-barrel channels of the bacterial outer membrane. There is no genetic connection between the mitochondrial beta-barrel VDACs and bacterial porins.

The review is well-structured and illustrated but it needs a vigorous editing of English grammar – some sentences are incomprehensible.  There are too many typos to list them. In the existing form, it cannot be published.

I appreciate that the reviewer considers me as an expert into the study of mitochondrial porins/VDAC. Indeed, we started with colleagues more than 40 years ago the study of these pores in lipid bilayer membranes and identified the first mammalian type in rat liver mitochondria. Because of the endosymbiotic theory we termed the 30 kDa protein mitochondrial porin, although we had to admit that the term VDAC existed already for pores from Paramecium mitochondria. Since then, both description of the mitochondrial pore existed and are well established in the literature. Andrei Lupaz and Doran Rapoport studied the evolution of mitochondrial porins/VDACs and concluded that they are built from β-strands of bacterial origin. I think that we should keep both terms in the manuscript.

It is correct that I described the isolation and purification of porin/VDAC in some detail. In the revised version it was somewhat shortened. On the other hand, it is not necessarily possible to bring in bacteria expressed eukaryotic proteins in a native form. There may exist examples where mitochondrial porins/VDACs expressed in bacteria do not adopt the native form. In these cases, the proteins have to be isolated from mitochondria with methods that are described in section 2. of the paper.

I have a few more comments:

  1. I appreciate the fact that the literature about VDAC is enormous and has kept rapidly growing since 2008 when the VDAC1 3D structure was solved, thus making a selection of what to cite and how to focus a review a serious challenge for any author. However, I cannot agree with the Author’s choice to cite almost exclusively his own works in Chapter 5 on the electrophysiology of mitochondrial porins and almost ignore Colombini’s group’s works on VDAC channel biophysics. Here are just a few examples:

I respectfully disagree that the work of Dr. Colombini was not adequately cited in the manuscript. The first version contained already 25 citations of his work because of his important contributions to this topic. I added the paper by Tan and Colombini (2007) in addition to the reference list of the revised manuscript. In this respect, I would like to refer to the review of Dr. Colombini (Colombini M. The VDAC channel: Molecular basis for selectivity. Biochim Biophys Acta. 2016 1863(10):2498-502). The review contains 35 citations in total, 25 refer to the work of Dr. Colombini. The name Benz is missing in the reference list.

-       Page 11: “…presumably because the selectivity of the pore in the open and closed configuration is different.” This is not a “presumable” situation” - the selectivity of open and closed VDAC states has been experimentally measured by Colombini (J Membr Biol 1989) in the KCl gradient. These are published data.

This sentence was corrected, and reference was given to Tan and Colombini (2007).

-       Page 15: “The ion selectivity of the closed states cannot be measured using zero-current membrane potential measurements with pores in the closed state because an external voltage must be applied to close the pores.” This is simply incorrect. Please read Colombini’s works on VDAC selectivity and how to measure it. See e.g., Tan & Colombini BBA, 2007.

This sentence was changed, and reference was given to Tan and Colombini (2007).

-       Page 13: “This result suggested again that the number of gating charges involved in channel gating is about two”. This is incorrect: VDAC gating charge was experimentally measured by Colombini and coauthors and by other labs and is ~ 3e. This is based on many different publications on VDAC channel gating.

On page 19, I mentioned that the number of gating charges varied for different studies.

  1. Chapter 6: “at the cis-side the channel was always in its open configuration even for voltages up to 120 mV and higher”. Does the Author have any explanation for such an unusual VDAC behavior in the presence of the polyanion? The following statement regarding bacterial porins that “the polyanion can interact with the gate from both sides of the channel” seems to contradict the effect of the polyanion on VDAC gating (Fig. 5).

It seems that the gate within the mitochondrial porins/VDACs is kept in the open position, when polyanion binds to the pore. Figure 5 shows the addition of polyanion to only one side of the pore. When it is added to both sides of the pore, the response of gating to voltage is symmetrical.

-       It is not clear why the Author thinks that “a direct polyanion-induced block of the channels is rather unlikely”?  What is the experimental evidence for such a conclusion? On page 23 the Author compares, quite rightly, the interaction of polyanion and VDAC with polyanionic C-terminus of tubulin, which is a permeation block.

A polyanion-induced block is relatively unlikely because the conductance of the voltage-induced closed state of porin/VDAC is very similar to that of the polyanion-induced closed state. This means that polyanion interacts with the gate and does not directly block the pore.

-       Why is the increase of cation selectivity in the presence of polyanion surprising if the polyanion-induced closed VDAC state is always more cation-selective and selectivity depends on the type of anion and its concentration? This has been described in a few works by different groups. See, e.g., a work on VDAC selectivity (Zambrowicz & Colombini BJ 1993) and OmpF selectivity (Alcaraz et al BJ 2009 doi:10.1016/j.bpj.2008.09.024).

Reviewer #1 is correct, the increase in cation selectivity is not surprising in presence of the polyanion. The corresponding sentence was corrected.

  1. Part 6.1: the way it is written gives the impression that the effect of the polyanion on mitochondrial adenylate and creatine kinases is the only reliable result indicating that ATP crosses the outer membrane through VDAC. Maybe the Author did not mean it, but that is how this paragraph sounds. I thought that the translocation of ATP through the VDAC open state and the lack of translocation through the closed state are well-established facts demonstrated experimentally by Colombini’s group in 1996 (see e.g., doi: 10.1074/jbc.271.45.28006).

Reviewer #1 is correct, the translocation of ATP through the open state of porin/VDAC and its block through the closed state are well-established facts. This is mentioned in the corresponding paragraph.

  1. Chapter 6.1, page 18: in the discussion about “mitochondrial porin could be involved in the control of mitochondrial metabolism via its voltage-dependence” the theoretical works of Lemeshko on the role of VDAC-hexokinase complex in the generation of the potential across the outer membrane seem worth mentioning in the context of a general discussion of this paper.

The theoretical works of Lemeshko VV was cited in the revised manuscript, ref. [168].

  1. Chapter 7: The similarities between VDAC1 and OmpF structures are quite exaggerated in this chapter. The main similarity between them is that they both are both beta-barrel channels. The number of beta-strands is different, the construction of the constriction zone – alpha-helical N-terminus for VDAC and the loop for OmpF– are different, and the conductance of each monomer is different – 4 nS for VDAC and 1.4 nS for OmpF monomer in 1 M KCl. The comparison of monomer VDAC conductance with trimeric OmpF is not adequate.

The conductance of the OmpF monomer is now compared to that of the porin/VDAC monomer.

The references for Tom40 structure should be updated with the latest publications: Tucker and Park, Nature Struct Mol Biol 2019, and Araiso et al Nature 2019.

I agree that it is worthwhile to cite here a more recent reference for Tom40 structure and selected Araiso et al. Annu Rev Biochem. 2022 91 : 679-703. [73]

  1. Could the statement on Page 20 “the fact that bacterial outer membrane pores have only passive properties, whereas mitochondrial porins adopted during evolution an active role in mitochondrial metabolism” be clarified? Bacterial and mitochondrial porins form passive diffusion pores. Their physiological roles are different.

The role of mitochondrial porins versus that of bacterial porins were clarified on Page 20.

  1. Part 7.1, page 21: I believe that it should be deletion of 1-31 residues in VDAC2 in Gattin et al, 2015.

Reviewer #1 is correct. Amino acids 1 to 31 were removed in VDAC2. This was corrected in the revised manuscript.

  1. Chapter 8: The statement “The voltage-induced closed states of mitochondrial porins (Benz et al., 1990) favor the transport of cations, in particular of Ca2+” needs a reference  to Tan&Colombini BBA 2007, who showed that VDAC closed states are significantly more permeable for calcium than the open state.

Reference to Tan and Colombini BBA 2007 was added.

Comments on the Quality of English Language

The paper needs thorough proofreading for English grammar and typos before it can be published in your Journal. 

Proofreading was done.

Reviewer 2 Report

Comments and Suggestions for Authors

This review describes current knowledge of the anionic voltage-dependent channel (VDAC). The author of this review has been working on this protein for many years and therefore has in-depth knowledge of the subject. He emphasises the role of this mitochondrial porin, other than that of a passive filter, in contrast to vacterial porins.

Consequently, I recommend this manuscript for publication. 

You will find below my more detailed comments.

Comments:

1.     Although the abstract gives interesting information about the origin of mitochondria, this information should, in my opinion, be in the introduction. The abstract should be rewritten focusing more on the content of the review.

2.     Model GHK 

Is it still appropriate to use the Goldmann-Hodgkin-Katz model? There have been other models that can better estimate ion selectivity through VDAC. Can the author comment on this aspect?

3.     The section "Isolation and purification of eukaryotic porins" is written in a rather technical manner and is not very pleasant to read. However, it must be difficult to change the type of writing given the subject matter.

4.     Figures 8 and 9 depicting the 3D structures of VDAC and the location of important amino acids are of rather poor quality. A program such as vmd or pymol can easily produce better quality figures.

5.     Functional residues

Section "7.1. Functional amino acids in the 3D structure of mitochondrial porins" discusses only the "GLK" triplet but other conserved amino acids have been shown to be important (e.g. D15, K19, K95, K236 and R252 (S. cerevisae numbering).

Comments on the Quality of English Language

Ok.

Author Response

This review describes current knowledge of the anionic voltage-dependent channel (VDAC). The author of this review has been working on this protein for many years and therefore has in-depth knowledge of the subject. He emphasises the role of this mitochondrial porin, other than that of a passive filter, in contrast to vacterial porins.

Consequently, I recommend this manuscript for publication. 

I appreciate the positive view of reviewer #2 on the manuscript.

You will find below my more detailed comments.

 Comments:

  1. Although the abstract gives interesting information about the origin of mitochondria, this information should, in my opinion, be in the introduction. The abstract should be rewritten focusing more on the content of the review.

Reviewer #2 is correct. The abstract was rewritten and the information about the origin of mitochondria was transferred to the introduction section.

  1. Model GHK 

Is it still appropriate to use the Goldmann-Hodgkin-Katz model? There have been other models that can better estimate ion selectivity through VDAC. Can the author comment on this aspect?

The GHK model provides still the central equation to calculate the ion selectivity of a pore or a channel because it assumes a constant field within the pore. Even when the current of an anion or a cation through a pore is considered, the current of single ions is given by the GHK current equation. This is explicitly shown by eqs. (1) to (3) in the paper by Tan and Colombini (2007) and in many other studies.

  1. The section "Isolation and purification of eukaryotic porins" is written in a rather technical manner and is not very pleasant to read. However, it must be difficult to change the type of writing given the subject matter.

I tried to modify the corresponding section in the revised manuscript.

  1. Figures 8 and 9 depicting the 3D structures of VDAC and the location of important amino acids are of rather poor quality. A program such as vmd or pymol can easily produce better quality figures.

I tried to improve figures 8 and 9 with the programs suggested by reviewer #2 but it was not possible to improve them in the short time I had.

  1. Functional residues

Section "7.1. Functional amino acids in the 3D structure of mitochondrial porins" discusses only the "GLK" triplet but other conserved amino acids have been shown to be important (e.g. D15, K19, K95, K236 and R252 (S. cerevisae numbering).

The section 7.1 does not discuss only the GLK triplet but also E73, K12, L10 and V143. K95 is part of the GLK triplet. K236 and R252 are only preserved in some sequences and not throughout all mitochondrial porins. Some of the preserved residues are now mentioned in the revised version of the manuscript on page 28.

Reviewer 3 Report

Comments and Suggestions for Authors

This is an excellent review article on mitochondrial porins written by a leading expert of the field. The article discusses the development and current status of the field in a convincing and balanced manner. It will be an excellent resource for the scientific community, covering mitochondrial porins and their functions in vitro and in vivo. I fully support publication of this article.

I have two minor comments:

1. Abstract, second sentence: The symbiosis occurred about 1.5 billion years ago (not: million!).

2. The first sentence of the Introduction ('The knowledge of …') sounds similar to a sentence in previous papers of the author. Therefore this sentence should be re-worded. I recommend the author to also check if other sentences are similar to text in his previous articles.

Comments on the Quality of English Language

V

Author Response

This is an excellent review article on mitochondrial porins written by a leading expert of the field. The article discusses the development and current status of the field in a convincing and balanced manner. It will be an excellent resource for the scientific community, covering mitochondrial porins and their functions in vitro and in vivo. I fully support publication of this article.

I appreciate that reviewer #3 has such a positive response to the manuscript.

I have two minor comments:

  1. Abstract, second sentence: The symbiosis occurred about 1.5 billion years ago (not: million!).

This was corrected in the revised version of the manuscript.

  1. The first sentence of the Introduction ('The knowledge of …') sounds similar to a sentence in previous papers of the author. Therefore this sentence should be re-worded. I recommend the author to also check if other sentences are similar to text in his previous articles.

Overlap between this manuscript and previous articles were checked in the revised version.

Reviewer 4 Report

Comments and Suggestions for Authors

This is valuable review focusing on history of VDAC studies including sophisticated issue of electrophysiology. However, the review is strongly based on the review published in 2021 in Frontiers in Physiology (10.3389/fphys.2021.734226) which is reflected in copy/paste of some parts including some figures and tables together with their captions. I think that the references of relevant papers as well as the permission for usage of already published graphics should be added.

Moreover I would recommend careful proofreading as there are a few terminological and language problems in the review.

Some examples:

Terminology:

translation instead of translocation

Tom40 complex

Tob44/Sam50 protein

human isoforms of human porin

porin sequences 1 of Porin 31HL

respiratory substrates like glycolytic ATP, phosphate, and small cations

respiratory molecules

Some sentences:

Mitochondrial proteins do not exist only in eukaryotic cells.

Mitochondrial porins are similar as most mitochondrial proteins encoded by the nucleus, synthesized at cytoplasmic ribosomes and transported post-translationally into mitochondria.

The 3D-structures of mVDAC1 are shown from the side in direction to the N- and C-termini of the protein (probably to the surface of the mitochondrion) and the surface of the bacterial cell (upper structures) and from the N- and C-termini of mVDAC1 and the surface of the bacterial cell (structures down).

18 ß-strands are antiparallel like the situation in most bacterial porins. ß-strands one and nineteen are in a parallel configuration – the issue of the strands’ number is more complex.

Moreover the term “plastid porins” should be explained

I would also suggest Conclusion rewriting because at present form it is rather a summary of the review. Additionally, I would suggest consideration of cysteine residues in part 7 “Structure of the mitochondrial outer membrane pore”; e.g.,  10.3389/fphys.2021.750627.

Author Response

This is valuable review focusing on history of VDAC studies including sophisticated issue of electrophysiology. However, the review is strongly based on the review published in 2021 in Frontiers in Physiology (10.3389/fphys.2021.734226) which is reflected in copy/paste of some parts including some figures and tables together with their captions. I think that the references of relevant papers as well as the permission for usage of already published graphics should be added.

Reviewer #4 is correct. Some of the Tables were taken from the Frontiers of Physiology review. However, the data were taken from the published literature and credit was given to the papers, where the data were originally published.

Moreover I would recommend careful proofreading as there are a few terminological and language problems in the review.

Some examples:

Terminology:

translation instead of translocation

Tom40 complex

Tob44/Sam50 protein

human isoforms of human porin

porin sequences 1 of Porin 31HL

respiratory substrates like glycolytic ATP, phosphate, and small cations

respiratory molecules

The manuscript was subjected to careful proofreading to avoid terminological and language problems.

Some sentences:

Mitochondrial proteins do not exist only in eukaryotic cells.

Mitochondrial porins are similar as most mitochondrial proteins encoded by the nucleus, synthesized at cytoplasmic ribosomes and transported post-translationally into mitochondria.

The 3D-structures of mVDAC1 are shown from the side in direction to the N- and C-termini of the protein (probably to the surface of the mitochondrion) and the surface of the bacterial cell (upper structures) and from the N- and C-termini of mVDAC1 and the surface of the bacterial cell (structures down).

18 ß-strands are antiparallel like the situation in most bacterial porins. ß-strands one and nineteen are in a parallel configuration – the issue of the strands’ number is more complex.

The sentences were checked, and changes were introduced where appropriate.

Moreover the term “plastid porins” should be explained

The term plastid porin was explained in the revised manuscript on page 7.

I would also suggest Conclusion rewriting because at present form it is rather a summary of the review. Additionally, I would suggest consideration of cysteine residues in part 7 “Structure of the mitochondrial outer membrane pore”; e.g.,  10.3389/fphys.2021.750627.

The section Conclusions was revised. The putative function of the cysteines within the primary sequence of porin 31HL (hVDAC1) was discussed in section 7 of the revised manuscript.